# Profiling Plasma Cytokines by A CRISPR-ELISA Assay for Early Detection of Lung Cancer

**DOI:** 10.3390/jcm11236923

**Published:** 2022-11-24

**Authors:** Ning Li, Molangur Chinthalapally, Van K. Holden, Janaki Deepak, Pushpa Dhilipkannah, Jonathan M. Fan, Nevins W. Todd, Feng Jiang

**Affiliations:** 1Departments of Pathology, University of Maryland School of Medicine, Baltimore, MD 21201, USA; 2Environmental Science and Technology, College of Agriculture and Natural Resources, University of Maryland, College Park, MD 20742, USA; 3Departments of Medicine, University of Maryland School of Medicine, Baltimore, MD 21201, USA; 4Montgomery Blair High School, Silver Spring, MD 20901, USA

**Keywords:** lung cancer, cytokines, CRISPR, ELISA, biomarkers

## Abstract

Cytokines play crucial roles in tumorigenesis and are potential biomarkers for cancer diagnosis. An Enzyme-linked Immunosorbent Assay (ELISA) is commonly used to measure cytokines but has a low sensitivity and can only detect a single target at a time. CRISPR-Associated Proteins (Cas) can ultra-sensitively and specifically detect nucleic acids and is revolutionizing molecular diagnostics. Here, we design a microplate-based CRISPR-ELISA assay to simultaneously profile multiple cytokines, in which antibodies are coupled with ssDNA to form antibody-ssDNA complexes that bridges CRISPR/Cas12a and ELISA reactions. The ssDNA triggers the Cas12a collateral cleavage activity and releases the fluorescent reporters to generate amplified fluorescent signals in the ELISA detection of cytokines. The CRISPR-ELISA assay can simultaneously measure multiple cytokines with a significantly higher sensitivity compared with conventional ELISA. Using the CRISPR-ELISA assay to profile plasma cytokines in 127 lung cancer patients and 125 cancer-free smokers, we develop a panel of plasma cytokine biomarkers (IL-6, IL-8, and IL-10) for early detection of the disease, with 80.6% sensitivity and 82.0% specificity. The CRISPR-ELISA assay may provide a new approach to the discovery of cytokine biomarkers for early lung cancer detection.

## 1. Introduction

Lung cancer is the leading cause of cancer-related deaths in the USA and worldwide. Over 85% of lung tumors are non-small cell lung cancers (NSCLCs). NSCLC is a heterogenous disease, which mainly consists of two histological types: squamous cell carcinoma (SCC) and adenocarcinomas (AC). Low-dose CT (LDCT) screening for the early detection of lung cancer leads to a 20% relative reduction in mortality from the disease as compared to chest X-rays [1]. Therefore, LDCT is recommended to be used for early lung cancer detection [2]. However, LDCT for the early detection of NSCLC is associated with overdiagnosis, excessive cost, and radiation exposure. The development of circulating biomarkers that can sensitively and specifically diagnose early-stage lung cancer is required.

Chronic inflammation and immune responses contribute to the development and progression of lung cancer [3,4]. Cytokines are key signaling molecules in the inflammation and immune systems and are produced by surrounding cells or malignant cells as part of the inflammatory process and hypoxic conditions that accompany tumor growth, invasion, and metastasis [3,4]. Therefore, profiling cytokines could aid in the early detection, prognostication, and monitoring of the therapeutic responses and outcomes of human tumors. For instance, elevated serum IL-6 and IL-8 levels were associated with shorter survival in patients with lung cancer [5]. The measurements of plasma IL-9, Eotaxin, G-CSF, and TNF-α have potential for the diagnosis of colon cancer [6]. Therefore, the detection of cytokines in plasma or serum might provide novel biomarkers for the early detection of malignancies, including lung cancer.

Enzyme-linked Immunosorbent Assay (ELISA) is considered the gold standard of immunoassays and is widely used to measure cytokines [7]. However, ELISA has a low sensitivity due to the nonspecific binding of horseradish peroxidase conjugates and the associated high backgrounds [7]. This can be particularly problematic for the use of ELISA in the clinical settings since abundances of cytokines in serum or plasma are usually low. A platform that can more sensitively perform cytokine profiling is needed.

Clustered regularly interspaced short palindromic repeats (CRISPRs) are DNA sequences within the genomes of prokaryotic organisms [8,9]. The CRISPR-associated immune system can cleave specific nucleic acid sequences and is now used in gene editing. New evidence shows that CRISPR-Associated Proteins (Cas) can unleash nonspecific endoribonuclease activity to degrade DNA and RNA, and hence enable the ultra-sensitive and specific detection of nucleic acids [8,9]. Therefore, CRISPR-Cas is revolutionizing molecular diagnostics [8,9]. Using Cas12a and Cas13a, Chen et al. and Gootenberg et al. developed DETECTR and SHERLOCK for the detection of viruses such as HPV, ZIKV, and DENV [8,9]. We have shown that CRISPR-Cas12a can detect HPV, SARS-CoV-2, influenza, and respiratory syncytial viruses, and druggable genomic DNA mutations with a higher sensitivity compared with PCR and DNA sequencing [10,11,12,13]. We have also developed a microplate-based CRISPR assay to simultaneously and rapidly detect multiple molecular targets [14]. Lee et al. [15] and Li et al. [16] recently demonstrated that the Cas12a-based trans-cleavage activity of fluorescently quenched (FQ) reporter substrates could improve the sensitivity of ELISA for protein detection. However, the techniques only detect a single target at a time, restricting their applications in the development of multiple biomarkers for more precise diagnosis of lung cancer. Herein, we combine our microplate-based CRISPR assay [14] and ELISA to develop a high-throughput platform that can perform multiple cytokine profilings at once. Our results shows that the CRISPR-ELISA assay has a higher analytic sensitivity for the simultaneous detection of multiple cytokines compared with the conventional ELISA. Furthermore, using the CRISPR-ELISA assay to profile multiple cytokines in the plasma of lung cancer patients and cancer-free smokers, we developed a panel of plasma cytokine biomarkers for the early detection of lung cancer with 80.6% sensitivity and 82.0% specificity.

## 2. Materials and Methods

***Patients and clinical information.*** This study was approved by the Institutional Review Boards of University of Maryland Baltimore (IRB HP-00040666). We consented and recruited lung cancer patients and cancer-free smokers in University of Maryland Medical Center using the inclusion and/or exclusion criteria recommended by U.S. Preventive Services Task Force for lung cancer screening [2]. eW recruited smokers between the ages of 55 and 80 who had at least a 30 pack–year smoking history and were former smokers (quit within 15 years). Exclusion criteria included age < 21 years, pregnancy or lactation, current pulmonary infection, thoracic surgery within 6 months, and radiotherapy to the chest within 1 year. The medical records were reviewed for their demographic, radiological, and clinical variables. A total of 127 NSCLC patients and 125 cancer-free smokers were recruited. Among the patients with lung cancer, 51 were female and 76 were male, 64 had stage I NSCLC, 26 with stage II, 23 with stage III, and 14 with stage IV. 127 NSCLC tumors consisted of 58 SCC and 69 AC, two major histological subtypes. Of the cancer-free smokers, 151 patients were female and 74 were male. The cases and controls were randomly grouped into two cohorts: a development cohort and a validation cohort. The development cohort consisted of 68 lung cancer patients and 69 cancer-free smokers, while the validation cohort comprised 59 lung cancer patients and 56 cancer-free smokers. The demographic and clinical variables of the two cohorts are shown in Table 1 and Table 2.

***Blood collection and plasma preparation***. From each participant, we collected 10 mL of whole blood in BD Vacutainer spray-coated K2EDTA Tubes (BD, Franklin Lakes, NJ) using the standard operating protocol, as described in our previous studies [17,18,19,20,21]. The specimens were processed within one hour of the collection by centrifugation at 1000× *g* for 15 min at 4 °C. Plasma was then aliquoted, frozen, and stored at –80 °C.

***Antibodies and recombinant human proteins.*** Human antibodies against 12 cytokines and the corresponding recombinant human proteins were obtained from Abcam (Abcam, Waltham, MA). The 12 cytokines include IL-1b, IL-2, IL-6, IL-7, IL-8, IL-10, IL-12p70, IL-13, IL-17A, MCP1, IFN-γ, and TNF-α.

***Designing a CRISPR-ELISA assay for analysis of multiple cytokines.*** To improve ELISA for the detection of cytokines by CRISPR, we first generated antibody-ssDNA (Abs-ssDNA) complex via streptavidin-biotin binding by using the streptavidin conjugation kit (Abcam), as previously described by Lee et al. and Li et al. [15,16] (Figure 1). The biotinylated ssDNA (GAA GAC ACC CTA CCA ACC CCC CCC TAA) [15,16] was synthesized by Integrated DNA Technologies (IDT, Redwood City, CA). The ssDNA was designed with a complementary sequence to the gRNA of CRISPR/Cas12a complex, and this triggered the Cas12a collateral cleavage activity. Abs-ssDNA complex was responsible for linking CRISPR/Cas12a activity and ELISA detection (Figure 2). In the procedure, the antibody was bound to the target analyte, while the ssDNA oligonucleotide was recognized by gRNA, which activated Cas12a collateral cleavage activity. The activated cleavage activity subsequently unquenched the fluorescent quenched (FQ) ssDNA reporters to produce amplified fluorescent signals that increase the sensitivity of ELISA detection (Figure 2). Second, we selected 12 cytokines for testing based on previous studies of the cytokines, whose expressions were associated with lung cancer [22,23]. The 12 cytokines include IL-1b, IL-2, IL-6, IL-7, IL-8, IL-10, IL-12p70, IL-13, IL-17A, MCP1, IFN-γ, and TNF-α. Based on our previous study [14], we designed a microplate-based CRISPR-ELISA assay that could specifically and simultaneously detect the 12 cytokines by using a 96-well microplate (Figure 2). Each target was tested in triplicate. First, 100 µL samples were added in each well. The plate was incubated for one hour at room temperature and washed with 1X PBS 3 times. Second, the Abs-ssDNA complexes were added to each well and acted as primary antibodies to recognize the corresponding cytokines. Third, 15 µL of the CRISPR/Cas12a reaction mixture (Cas12a, gRNA that complementarily match the ssDNA, and ssDNA-FQ reporter) was added to each well. Fourth, the CRISPR reaction was incubated at room temperature for another half hour, during which gRNA recognized the ssDNA and triggered Cas12a collateral cleavage activity. The activated CRISPR/Cas12a cleaved the reporters to produce fluorescent reporters and the released fluorescent reporters generated amplified fluorescent signals. Finally, fluorescent kinetics were then measured on a fluorescent plate reader for simultaneous detection of the multiple cytokines, as described in our previous studies [10,11,12,13,14].

***The measurement of the cytokines by conventional ELISA.*** ELISA immunoassays from Abcam were performed to measure IL-6 and TNF-α in parallel according to the manufacturer’s instruction and our previous study [24].

***Statistical analysis.*** We used the standard deviation of the response of the curve and the slope of the calibration curve to estimate the limit of detection (LOD). LOD = 3.3 σ/Slope, where: σ = the standard deviation of the response at low concentrations: Slope = the slope of the calibration curve [25]. We determined the reproducibility and precision of assays by using coefficient of variation (CV). CV values were accepted when they were less than 20%. We used linear regression to determine the correlation between different methods. Differences in values are evaluated using Student’s *t*-test. We used GraphPad Prism 9 software (Graphpad Inc; San Diego, CA, USA) to perform a statistical analysis and plot graphical presentation. We used univariate and multivariate analyses to determine which of plasma cytokines were associated with lung cancer. We also used Pearson’s correlation coefficient test to estimate the relationship of the plasma cytokines with demographic and clinical information of the cases and controls. To develop a panel of plasma cytokine biomarkers, we used a research design, as shown in Figure 3. First, the cytokines were analyzed using multivariate logistic regression models with constrained parameters as in least absolute shrinkage and selection operator (LASSO) based on receiver–operator characteristic (ROC) curves to identify an optimal panel of biomarkers in the development cohort. Second, the optimized biomarker panel was validated in the validation cohort through a comparison with final clinical diagnosis and the results in the development cohort using the area under the ROC curves (AUCs).

## 3. Results

### 3.1. CRISPR-ELISA Assay Can Sensitively Detect the Cytokines

First, we used two antibodies, anti-IL-6 and anti-TNF-α, to test various concentrations of the antibodies and ssDNA for preparing Abs-ssDNA complexes, as previously described by Li et al. and Lee et al. [15,16]. The molar ratio of antibody/ssDNA was optimized at 1:15, and the concentration of the Abs-ssDNA complex that was to be used in the reaction was optimized at 8 µg/mL. Furthermore, the optimized concentrations of LbaCas12a, gRNA, and ssDNA reporter were 5 µg/mL, 2.5 µg/mL, and 5µg/mL, respectively. With the concentrations, CRISPR-Cas12a with gRNA produced the highest fluorescence signals in ELISA detection of the cytokines, which significantly raised with an increase in reaction time. In addition, the system could produce significant fluorescence values as early as 10 min at room temperature. Moreover, when using the same optimal concentrations to prepare Abs-ssDNA complexes and perform ELISA and CRISPR, another 10 antibodies (IL-1b, IL-2, IL-7, IL-8, IL-10, IL-12p70, IL-13, IL-17A, MCP1, and IFN-γ) also produced the highest fluorescence signals to detect the respective cytokines. Therefore, the optimum protocol and concentrations of the reagents could be generally applicable to the detection of these cytokines.

Second, to determine the sensitivity of the CRISPR-ELISA assay to simultaneously detect the cytokines, the 12 analyte proteins were serially diluted in nuclease-free water (0, 1, 10, 100, 1,000, 10,000, and 100,000 pg/mL), respectively. Each diluted sample was applied to the CRISPR-ELISA procedures. For comparison, enzyme-linked immunoassays were performed to measure IL-6 and TNF-α in parallel. The conventional ELISA assays were chosen due to their sensitivity, robustness, and precision. As shown in Figure 4A, the limits of detection (LODs) of the CRISPR-ELISA assay and conventional ELISA assay for the detection of the cytokines were 16 and 166 pg/mL, respectively. Therefore, the CRISPR-ELISA assay could improve the analytic sensitivity of the conventional ELISA by 10-fold for the detection. Furthermore, to determine agreement between the CRISPR-ELISA assay and conventional ELISA, we used the two platforms to analyze IL-8 and TNF-α in the plasma of 20 lung-cancer patients. The results of the conventional ELISA assay in four and three samples were below the LOD of IL-6 and TNF-α, respectively. However, the results of the CRISPR-ELISA assay in all 20 samples were above the LOD of IL-6 and TNF-α. Linear regression analysis showed that the two platforms had a high agreement in the detection of the cytokines (R2 = 0.96 and 0.97, respectively; *p* < 0.01) (Figure 4B).

### 3.2. CRISPR-ELISA Assay Can Specifically and Reproducibly Detect the Different Cytokines

CRISPR-Cas12a complex with a specific antibody only exhibited positive results for the targeted cytokine, without cross-reactivity with others (Figure 4C), implying the high specificity of the CRISPR-ELISA assay to its target analytes. To determine reproducibility and precision, the diluted specimens were divided into three parts, and tested on days 1, 14 and 30, respectively. Furthermore, inter-assay precision was determined as the coefficient of variation (CV) of test samples at different dilutions. In addition, intra-assay precision was calculated by determining the CV over different wells on the same plate. The CRISPR-ELISA assay had CVs of 5.7–13.9% for quantification of the targets, indicating its high reproducibility and precision in detecting the cytokines (Appendix A).

### 3.3. Diagnostic Performance of the CRISPR-ELISA Assay for Analysis of the Cytokines in the Early Detection of Lung Cancer

We used the CRISPR-ELISA assay to profile the 12 cytokines in plasma of the development cohort of 68 NSCLC patients and 69 cancer-free subjects. Of the 12 cytokines, six (IL-6, IL-8, IL-10, IL-12p70, IFN-γ, and TNF-α) displayed a higher level in plasma of lung cancer patients compared with cancer-free smokers (All *p* < 0.001) (Table 3). The six individual cytokines showed AUC values of 0.67–0.76 when differentiating the cases from controls with 66.0–73.1% sensitivities and 60.1–76.0% specificities. We then used LASSO and AUCs to determine the performance of different patterns of combining the cytokines. Three cytokines (L-6, IL-8, and IL-10) were identified and optimized as a panel of biomarker, producing 0.79 AUC (Figure 5) (Table 4), which was statistically higher than that of any single one (*p* < 0.05). As a result, the biomarker panel could diagnose lung cancer with 80.6% sensitivity and 82.0% specificity. Furthermore, Pearson’s correlation coefficient test showed that there was no association of the biomarkers with the age and gender of the lung cancer patients and normal individuals (All *p* > 0.05). The biomarkers were associated with the smoking status of the cases and controls (*p* < 0.05). Interestingly, the biomarkers were not associated with the stages and histological types of lung cancer (All *p* > 0.05). Therefore, the plasma cytokine biomarker panel could be used for the diagnosis of an early-stage lung tumor with different histological types.

We further evaluated the diagnostic performance of the plasma cytokine biomarker panel in the validation cohort of 59 NSCLC patients and 56 controls. The three cytokines used together yielded 0.78 AUC. There was no statistical difference in the AUCs of the biomarker panel in the validation cohort and the development cohort (*p* = 0.46) (Figure 5) (Table 4). Furthermore, the biomarker panel could diagnose NSCLC with 78.3% sensitivity and 80.7% specificity in the validation cohort, confirming the diagnostic value. Consistent with the findings in the development cohort, no association of the biomarkers with the age and gender of the cases and controls was observed in the validation cohort (All *p* > 0.05). Furthermore, the biomarkers were not associated with the stages and histological types of lung cancer (All *p* > 0.05). Therefore, the validation study suggests that the plasma cytokine biomarker panel could be used for the diagnosis of early-stage lung tumor, regardless of the histological types. Taken together, the results imply the diagnostic significance of the panel of plasma cytokine biomarkers for the early detection of lung cancer.

## 4. Discussion

Profiling cytokines could provide novel biomarkers for the diagnosis of malignancies. ELISA is the most widely used method of immunodetection for cytokines in plasma or serum samples; however, the sensitivity is disappointingly poor. This can be particularly problematic for the use of ELISA in the early detection of lung cancer, as the levels of the circulating cytokines associated with early stages of the disease are commonly low. Furthermore, cancer is a heterogeneous disease and develops from complex molecular aberrations. Multiple biomarkers used in combination, rather than one used alone, are needed for the precise diagnosis of lung cancer. Therefore, the conventional ELISA that only measures a single cytokine at a time, with a low sensitivity, has limitations in the development of multiple biomarkers for the accurate diagnosis of lung cancer at the early stage.

To address these challenges, we developed a microplate-based CRISPR-ELISA assay for the concurrent analysis of cytokines based on our, and other, previous, studies [10,11,12,13,14,15,16]. Our head-to-head comparison shows that the CRISPR-ELISA assay can simultaneously measure multiple cytokines with similar specificity and reproducibility to conventional ELISA. Notably, the CRISPR-ELISA assay improves the sensitivity of the conventional ELISA by 10-fold. Furthermore, the results of different cytokines generated from the same plate could be directly read and presented in the same format as the expression levels in each well. Therefore, the CRISPR-ELISA assay can simplify the detection of multiple cytokines and minimize hands-on time, and thus provide consistent results across users and plates. Moreover, although we only measure 12 cytokines, the arrangement of numbers of antibodies is adjustable, depending on the protein targets that are to be detected by this array. For instance, if each target is tested in triplicate, a composition of up to 32 cytokines can be simultaneously qualified. It is also possible to extend this approach using a lager microplate (e.g., 486-well plate) to simultaneously assess other protein targets, such as growth factors and soluble receptors, in other types of clinical samples, including serum, culture media, lysates, or body fluids. Therefore, this assay might represent a new approach for the effective and comprehensive analysis of cytokine signaling pathways and diagnosis of diseases.

The early detection of NSCLC can significantly reduce the mortality. NSCLC is a heterogenous disease that comprises two main histological types, AC and SCC. Molecular biomarkers that can be used for the noninvasive and precise diagnosis of lung cancer at the early stage, regardless of the histological type, are urgently needed. By using a microplate-based CRISPR-ELISA assay, we developed a panel of plasma cytokine biomarkers consisting of IL-6, IL-8, and IL-10. The combined use of the three cytokines yields a higher diagnostic performance compared with one used alone. Furthermore, since changes in the three plasma cytokines are independent of the stage and histological type of lung cancer, the cytokine biomarkers might be used to identify lung cancer at the early stages, independent of the histological types. Given that lung cancer is the number one cause of cancer-related mortality, further use of the biomarker panel for early detection in the clinical settings may provide a potential approach to reduce mortality.

We acknowledge that more efforts are required to convert this study into a clinical test that can be applied in the laboratory settings. First, preparing the reagents in multiple steps of the combined CRSIPR and ELISA strategy mostly relies on manual labor, which might cause variations in test results and bias and overfitting for the development of biomarkers. To address the challenge, we will use a liquid handling robotic system to prepare and load the reagents in a rapid and accurate manner. Second, in this study, we improve the sensitivity of conventional ELISA by 10-fold using a direct strategy, in which the Abs-ssDNA is directly applied to recognize the target analyte as a single antibody. We will further increase the analytic sensitivity by using an antibody–analyte sandwich, in which the Abs-ssDNA will be used to target the Fc region of the second antibody, as described by Li et al. [16]. We will also increase the number and density of Cas12a recognition sites attached to the antibody to enhance the detection sensitivity, as proposed by Lee et al. [15]. Third, the plasma cytokine biomarkers are developed in the existing and retrospective sample sets, which may produce selection bias and overfitting. To diminish the bias and overfitting, we will perform a pivotal evaluation of the predictive accuracy of the biomarkers using a large sample size. Fourth, the panel of three plasma cytokine biomarkers was developed from a limited number of biomarker candidates (12 cytokines) for the diagnosis of lung cancer. Although it shows promise, the developed biomarker panel, with 80.6% sensitivity and 82.0% specificity, may not provide sufficient diagnostic significance in the clinical settings. Up to 80 cytokines have been identified as being associated with inflammatory and immunological responses^28^. We will use the CRISPR-ELISA assay to analyze other cytokines or cancer-associated proteins to identify the additional biomarkers that can be combined with the current ones for a more sensitive and specific diagnosis of lung cancer.

## 5. Conclusions

In summary, by taking advantage of the ultra-sensitive and specific detection of Cas-12a in nucleic acids, we have developed a high-throughput platform to simultaneously and sensitively quantify multiple cytokines. We further identified a panel of plasma cytokine biomarkers for the diagnosis of early-stage lung cancer. Nonetheless, more research is needed to improve the performance of the platform and validate the biomarkers in a large multi-center clinical project before they could be adopted in routine clinical settings.

## Figures and Tables

**Figure 1 jcm-11-06923-f001:**
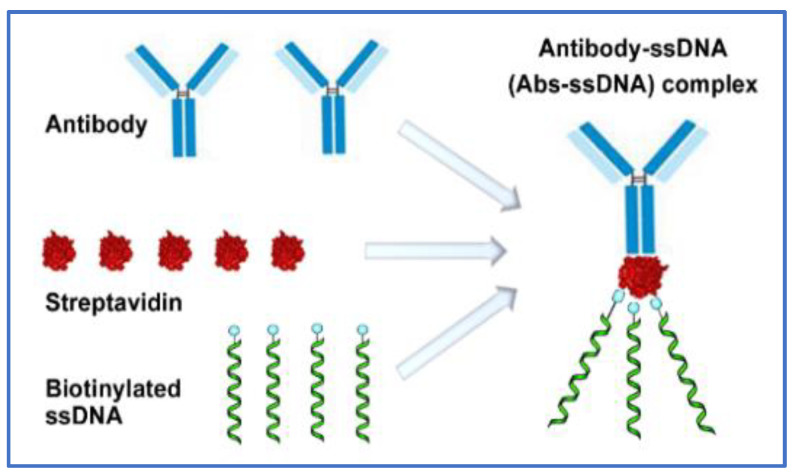
Generation of Abs-ssDNA complex. Through the streptavidin–biotin binding, the selected antibody is coupled with streptavidin and biotinylated ssDNA [15,16]. The created Abs-ssDNA complex can bridge CRISPR/Cas12a and ELISA reactions.

**Figure 2 jcm-11-06923-f002:**
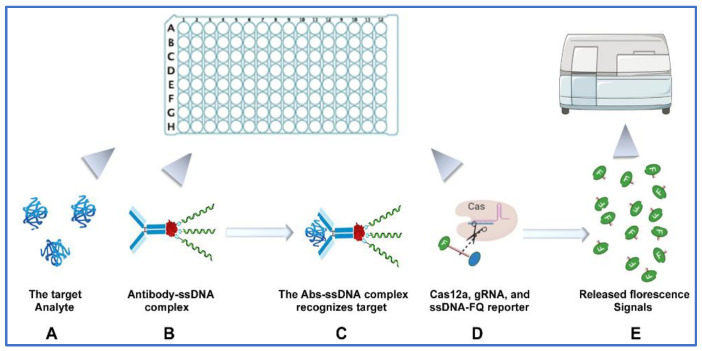
The workflow of CRISPR-ELISA assay to measure multiple cytokines. (**A**) Protein analytes were added to each well of a 96-well microplate. (**B**) The Abs-ssDNA complex (please see Figure 1) was loaded and functioned as primary antibody in ELISA. (**C**) The targeted cytokine in the protein samples was recognized by the Abs-ssDNA complex via the antibody. (**D**) Addition of CRISPR/Cas12a reaction mixture, which includes gRNA, was used to recognize the ssDNA and activate Cas12a collateral cleavage activity. Activation of CRISPR/Cas12a complex cut the reporters to produce fluorescent reporters. (**E**) The fluorescent reporters were detected by a fluorescent reader to reveal the concentration of target analytes.

**Figure 3 jcm-11-06923-f003:**
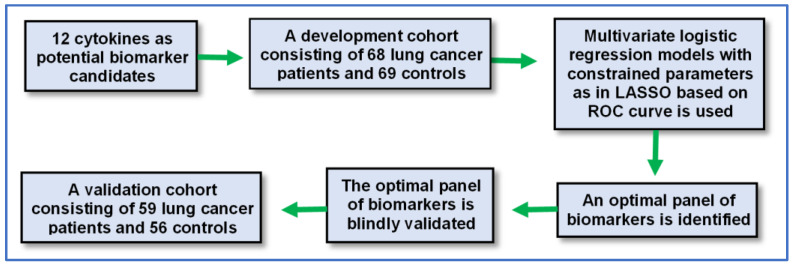
Diagram of the study design and patient flow. The 12 plasma cytokines were analyzed in a discovery cohort. Data were analyzed using multi-variate logistic regression models with constrained parameters, as in least absolute shrinkage and selection operator (LASSO), based on receiver–operator characteristic (ROC) curve to identify an optimal panel of biomarkers. The identified biomarker panel was then validated in the validation cohort.

**Figure 4 jcm-11-06923-f004:**
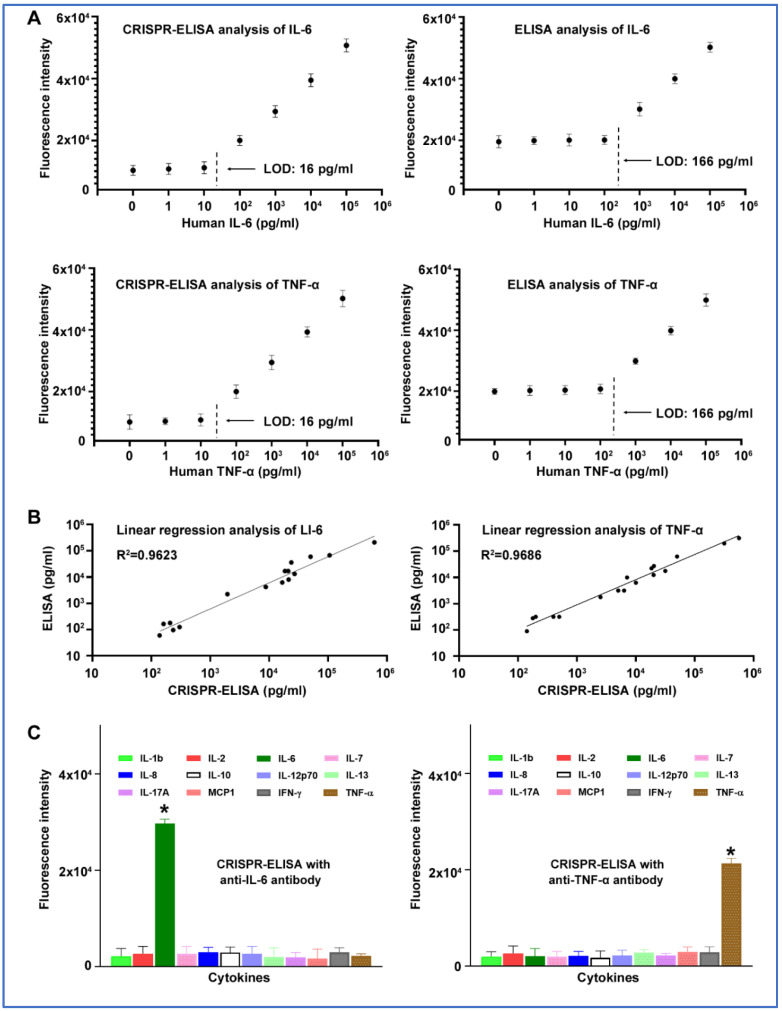
Analytic performance of the CRISPR-ELISA assay. (**A**) The analytical sensitivity of the CRISPR-ELISA assay and conventional ELISA for the detection of IL-6 and TNF-α in serially diluted standard samples. Each concentration of the serially diluted sample was tested in triplicates for three times. (**B**) Consistency between the CRISPR-ELISA assay and conventional ELISA for the detection of IL-6 and TNF-α. A total of 20 plasma samples of lung cancer patients were evaluated by the two platforms in parallel. The results were compared using linear regression analysis. (**C**) The analytic specificity of the CRISPR-ELISA assay for detection of cytokines was determined in a panel of human cytokines. The results of the CRISPR-ELISA assay, represented by fluorescence intensity, were read by a fluorescence plate reader in 10 min. The *X*-axis shows the sample of each cytokine. The *Y*-axis indicates the fluorescence intensity of each sample. The error bars represent the standard deviation from the mean of the fluorescence intensity generated from triplicates per sample. * *p* < 0.0001.

**Figure 5 jcm-11-06923-f005:**
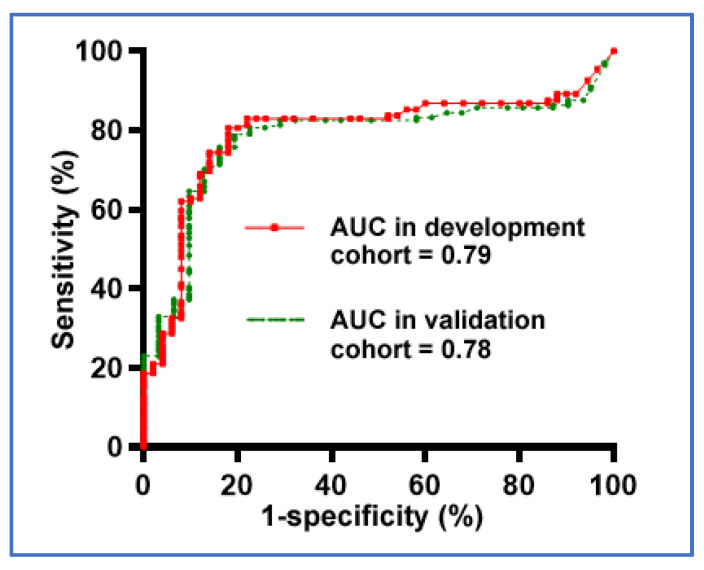
Diagnostic performance of the panel of three plasma cytokine biomarkers (IL-6, IL-8, and IL-10) for detection of lung cancer. ROC curve analysis is used to determine the diagnostic values. Red line indicates AUC value of the biomarker panel in the development cohort, while green line shows AUC value of the biomarker panel in the validation cohort. The AUC for each approach conveys its accuracy for diagnosis of lung cancer. There is no statistical difference between the two AUCs (*p* = 0.46).

**Table 1 jcm-11-06923-t001:** Characteristics of NSCLC patients and cancer-free smokers in a development cohort.

	NSCLC Cases (*n* = 68)	Controls (*n* = 69)	*p*-Value
Age	64.58 (SD 10.16)	60.34 (SD 12.18)	0.12
Sex			0.33
Female	27	28	
Male	41	41	
Smoking pack-years (median)	32.8	31.6	0.27
Stage			
Stage I	34		
Stage II	15		
Stage III	11		
Stage IV	8		
Histological type			
Adenocarcinoma	37		
Squamous cell carcinoma	31		

Abbreviations: NSCLC, non-small cell lung cancer. SD, standard deviation.

**Table 2 jcm-11-06923-t002:** Characteristics of NSCLC patients and cancer-free smokers in a validation cohort.

	NSCLC Cases (*n* = 59)	Controls (*n* = 56)	*p*-Value
Age	64.29 (SD 9.37)	61.48 (SD 11.35)	0.15
Sex			0.38
Female	24	23	
Male	35	33	
Smoking pack-years (median)	32.8	31.6	0.23
Stage			
Stage I	30		
Stage II	11		
Stage III	12		
Stage IV	6		
Histological type			
Adenocarcinoma	32		
Squamous cell carcinoma	27		

Abbreviations: NSCLC, non-small cell lung cancer. SD, standard deviation.

**Table 3 jcm-11-06923-t003:** Diagnostic performance of six plasma cytokines for lung cancer in a development cohort.

Cytokines	*p*-Value	AUC (95% CI)	Sensitivity (%) (95% CI)	Specificity (%) (95% CI)
IL-6	<0.0001	0.6726 (0.6102 to 0.7349)	67.55 (59.46% to 74.93%)	60.13 (51.91% to 67.95%)
IL-8	0.0002	0.7174 (0.6166 to 0.8182)	66.00 (51.23% to 78.79%)	68.00 (53.30% to 80.48%)
IL-10	0.0001	0.7179 (0.6156 to 0.8202)	67.31 (52.89% to 79.67%)	76.00 (61.83% to 86.94%)
IL-12p70	<0.0001	0.7650 (0.6688 to 0.8612)	72.00 (57.51% to 83.77%)	74.00 (59.66% to 85.37%)
IFN-γ	<0.0001	0.7394 (0.6418 to 0.8370)	73.08 (58.98% to 84.43%)	68.00 (53.30% to 80.48%)
TNF-α	<0.0001	0.7499 (0.6488 to 0.8510)	69.81 (55.66% to 81.66%)	73.33 (58.06% to 85.40%)

AUC, the area under receiver operating characteristic curve; CI, confidence interval.

**Table 4 jcm-11-06923-t004:** Diagnostic performance of the panel of three plasma cytokine biomarkers for lung cancer in the two cohorts.

	The Development Cohort	The Validation Cohort
*p*-value	<0.0001	<0.0001
AUC (95% CI)	0.7924 (0.7209 to 0.8639)	0.7781 (0.7017 to 0.8545)
Sensitivity (%) (95% CI)	80.62.78 (72.74% to 87.05%)	78.26 (71.09% to 84.37%)
Specificity (%) (95% CI)	82.00 (68.56% to 91.42%)	80.65 (62.53% to 92.55%)

AUC, the area under receiver operating characteristic curve; CI, confidence interval.

## Data Availability

The data that support the findings of this study are available from the corresponding author upon a reasonable request.

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
