# Peer review of "Profiling Plasma Cytokines by A CRISPR-ELISA Assay for Early Detection of Lung Cancer"

_jcm, 2022, doi:10.3390/jcm11236923_

Round 1
Reviewer 1 Report
The authors conclude that CRISPR-ELISA assay may provide a new approach toward the discovery of cytokine biomarkers (IL-6, IL-8, and IL-10) for early lung cancer detection. The reviewer thinks this article is very interesting and has valuable information for cancer research, however, it needs to clear and complete in this article. Several corrections should be made:
1) Tables 1, 2 : NSCLC stage I = 34 and 30 cases (50%), respectively. The authors are requested to briefly discuss the of results between NSCLC stage I and other stages according to early detection of lung cancer.
2) Table 3: The authors are requested to discuss more about how to selected IL-6, IL-8, and IL-10 for further experiment. If any specific reason? It would be helpful if the authors give example or scenario to support its description.
Author Response
1) Tables 1, 2 : NSCLC stage I = 34 and 30 cases (50%), respectively. The authors are requested to briefly discuss the of results between NSCLC stage I and other stages according to early detection of lung cancer.
Response: As requested, we provide the discussion in both Result and Discussion Sections of the revised manuscript. The identified biomarkers were not associated with stages of lung cancer. Therefore, the plasma cytokine biomarker panel could be used for detection of both early-stage and advanced stage lung tumor with the same diagnostic values.
2) Table 3: The authors are requested to discuss more about how to select IL-6, IL-8, and IL-10 for further experiment. If any specific reason? It would be helpful if the authors give example or scenario to support its description.
Response: To address the comment, we add a new diagram (A new figure 3) to show how we have selected IL-6, IL-8, and IL-10 for further experiment. Essentially, the 12 plasma cytokines were tested in a discovery cohort. The generated data are analyzed by using multi-variate logistic regression models with constrained parameters as in least absolute shrinkage and selection operator (LASSO) based on receiver-operator characteristic (ROC) curve to identify and optimize a panel of biomarkers. From the development cohort, we identified three cytokines (L-6, IL-8, and IL-10), which used in combination produced a higher diagnostic value than that of any single one used alone. The three cytokines were then validated in a different cohort. The diagram and the associated description may provide scenario to support its description.
We thank the reviewer for providing the excellent comments for improvement of our manuscript.

Reviewer 2 Report
With all the limitation in imaging, there is need for novel biomarkers for diagnosis of malignancies. The ability to detect multiple biomarkers in very low concentrations are needed to aid in detecting early stages of cancers such as lung. The new methodology described in this paper could be a valuable instrument as detection levels for cytokines are much lower than conventional ELISA methods. I am no expert in the technical aspects of the study but the design is straight forward with a test and a validation group.
A couple of questions remain after reading the paper which appears well written and clear in terms of rationale:
The authors should be very clear that this study only has relevance for lung cancer and I think the authors should consider their plan to limit the panel from 12 to 3 cytokines as their original sample is small and lung cancers are very heterogeneous. A narrow panel might therefore lack in sensitivity.
The description of cancer and non cancer subjects included is very brief (non existing). There should be a reasonable description of how and where these subjects were recruited.
Author Response
The authors should be very clear that this study only has relevance for lung cancer, and I think the authors should consider their plan to limit the panel from 12 to 3 cytokines as their original sample is small and lung cancers are very heterogeneous. A narrow panel might therefore lack in sensitivity.
Response: We agree with the reviewer about original sample is small and the study only has relevance for lung cancer. As we discussed in the Discussion Section, we acknowledge that we will perform a prospective study using a large sample size to validate the biomarkers. To further address the comment, we provide more discussion to address the concern in the revised manuscript.
By using multi-variate logistic regression models with constrained parameters as in least absolute shrinkage and selection operator (LASSO) based on receiver-operator characteristic (ROC) curve to analyze the results, we successfully identified three cytokines (L-6, IL-8, and IL-10), which used in combination produced a higher diagnostic value than that of any single one. As shown in Tables 3-4, the panel of three biomarkers created better sensitivity and specificity than each of individual ones used alone.
We also agree with the reviewer about lung cancers are very heterogeneous, which mainly consists of AC and SCC. Furthermore, lung cancer develops from complex molecular aberrations through various mechanisms. Therefore, analysis of a single molecular biomarker might not provide enough diagnostic significance for lung cancer diagnosis. Interestingly, the panel of three plasma cytokine biomarkers could detect lung cancer regardless of stages and histological types of tumors. Therefore, the developed biomarker panel might overcome the challenge of diagnosis of lung cancer that is a heterogenous disease. To address the comment, we provide more discussion to address the concern in the revised manuscript.
The description of cancer and non-cancer subjects included is very brief (non existing). There should be a reasonable description of how and where these subjects were recruited.
Response: As requested, we provide more discussion about description of cancer and non-cancer subjects in the Method Section of the revised manuscript.
English language and style are fine/minor spell check were performed.
Finally, we thank the reviewer for providing the helpful suggestions for improvement of our manuscript.
